# Estimating Committor Functions via Deep Adaptive Sampling on Rare Transition Paths

## Abstract

The committor functions are central to investigating rare but important events in molecular simulations. It is known that computing the committor function suffers from the curse of dimensionality. Recently, using neural networks to estimate the committor function has gained attention due to its potential for high-dimensional problems. Training neural networks to approximate the committor function needs to sample transition data from straightforward simulations of rare events, which is very inefficient. The scarcity of transition data makes it challenging to approximate the committor function. To address this problem, we propose an efficient framework to generate data points in the transition state region that helps train neural networks to approximate the committor function. We design a Deep Adaptive Sampling method for TRansition paths (DASTR), where deep generative models are employed to generate samples to capture the information of transitions effectively. In particular, we treat a non-negative function in terms of the integrand in the loss functional as an unnormalized probability density function and approximate it with the deep generative model. The new samples from the deep generative model are located in the region of the transition and fewer samples are located in the other region, which provides effective samples for approximating the committor function and significantly improves the accuracy. We demonstrate the effectiveness of the proposed method with both simulations and realistic examples.

## 1 Introduction

Understanding transition events between metastates in a stochastic system plays a central role in chemical reactions and statistical physics (Okuyama-Yoshida et al., 1998; E & Vanden-Eijnden, 2006; Berteotti et al., 2009; E & Vanden-Eijnden, 2010). The physical process can be formulated as the following stochastic differential equation (SDE)

$$d\boldsymbol{X}_t = -\nabla V(\boldsymbol{X}_t)dt + \sqrt{2\beta^{-1}}d\boldsymbol{W}_t, \tag{1}$$

where $\boldsymbol{X}_t \in \Omega \subset \mathbb{R}^d$ is the state of the system at time $t$, $V : \Omega \mapsto \mathbb{R}$ denotes a potential function, $\beta$ is the inverse temperature, and $\boldsymbol{W}_t$ is the standard $d$-dimensional Wiener process. For two disjoint subsets of this stochastic system, we are interested in the transition rate, which can be characterized by the *commttor function*. For two distinct metastable regions $A, B \subset \Omega$, and $A \cap B = \emptyset$, denoting by $\tau_\omega$ the first hitting time of a subset $\omega \subset \Omega$ for a trajectory, the committor function $q : \Omega \mapsto [0, 1]$ is defined as $q(\boldsymbol{x}) = \mathbb{P}(\tau_B < \tau_A | \boldsymbol{X}_0 = \boldsymbol{x})$, where $\mathbb{P}$ denotes the probability. The committor function is a probability that a trajectory of SDE starting from $\boldsymbol{x} \in \Omega$ first reaches $B$ rather than $A$. By definition, it is easy to verify that $q(\boldsymbol{x}) = 0$ for $\boldsymbol{x} \in A$ and $q(\boldsymbol{x}) = 1$ for $\boldsymbol{x} \in B$. This committor function provides the information of process of a transition, and it is governed by the following partial differential equation (PDE) (Lai & Lu, 2018; Li et al., 2019)

$$\begin{aligned}
-\beta^{-1}\Delta q(\boldsymbol{x}) + \nabla V(\boldsymbol{x}) \cdot \nabla q(\boldsymbol{x}) &= 0, & \boldsymbol{x} &\in \Omega\backslash(A \cup B), \\
q(\boldsymbol{x}) &= 0, & \boldsymbol{x} &\in A, \\
q(\boldsymbol{x}) &= 1, & \boldsymbol{x} &\in B, \\
\nabla q(\boldsymbol{x}) \cdot \boldsymbol{n} &= 0, & \boldsymbol{x} &\in \partial\Omega\backslash(A \cup B),
\end{aligned} \tag{2}$$

where $\boldsymbol{n}$ is the outward unit normal vector of the boundary $\partial\Omega\backslash(A \cup B)$. Once the committor

function $q(\boldsymbol{x})$ is found, we can use it to extract the statistical information of reaction trajectories (E & Vanden-Eijnden, 2006; 2010).

Obtaining the committor function $q$ needs to solve the above high-dimensional PDE, which is computationally infeasible for traditional grid-based numerical methods. Some efforts have been made to employ deep neural networks to solve it (Khoo et al., 2019; Li et al., 2019; 2022). Training deep neural networks to approximate the committor function requires data points, which is usually achieved by sampling from the equilibrium distribution of the SDE. When the transition is rare, the samples from the transition state region are difficult to obtain from the SDE. As shown in the literature (Rotskoff et al., 2022; Kang et al., 2024), if the data points are not generated from the transition paths, then the trained neural network for approximating the committor function will have a large generalization error. To address this problem, we propose a new framework called Deep Adaptive Sampling on rare TRansition paths (DASTR) to train the deep neural network. More specifically, we generate samples in the transition region using an iterative construction. To do this, we define a proper sampling distribution using the current approximation of the committor function by neural networks and the potential function in the SDE. This sampling distribution that reveals the transition information is approximated by a deep generative model based on which new samples are generated and added to the training set. Once the training set is updated, the neural network model for the approximation of the committer function is further trained for refinement. This procedure is repeated to form the algorithm of deep adaptive sampling on rare transition paths. In other words, we push the samples into regions where the transition is by constructing a proper sampling distribution step by step. In this way, effective samples are selected to train the model, resulting in a better approximation of the committor function. The main contributions of this work are as follows.

- We propose a general framework, called deep adaptive sampling on rare transition paths, for estimating the high-dimensional committor function.

- We demonstrate the efficiency of the proposed method with numerical studies, including the alanine dipeptide problem.

## 2 RELATED WORK

We summarize the most related lines of this work.

**Neural Networks for Committor Functions.** Committor functions are represented by deep neural networks and can be trained by minimizing a variational loss functional. The training data points for discretizing the variational loss are usually sampled from the Gibbs measure (Khoo et al., 2019; Li et al., 2020; 2022), which needs to simulate the stochastic differential equations. This sampling method is inefficient due to the scarcity of transition data, especially for realistic systems at low temperatures. So, the committor function cannot be approximated well based on such a sampling strategy. Modified sampling methods are proposed in (Li et al., 2019; Rotskoff et al., 2022; Hasyim et al., 2022; Kang et al., 2024; Lin & Ren, 2024) to alleviate this issue, where a new probability measure for sampling is employed by modifying the potential function to produce enough data points in the transition region. Our approach generalizes these sampling strategies.

**Adaptive Sampling of Neural Network Solver.** The basic idea of adaptive sampling involves utilizing a non-negative error indicator, such as the residual square, to refine collocation points in the training set. Sampling approaches (Gao & Wang, 2023) (e.g., Markov Chain Monte Carlo) or deep generative models (Tang et al., 2023; Wang et al., 2024; Tang et al., 2024) are often invoked to sample from the distribution induced by the error indicator. Typically, an additional deep generative model (e.g., normalizing flow models) or a classical model (e.g., Gaussian mixture models (Gao et al., 2023; Jiao et al., 2023)) for sampling is required. This work uses the variational formulation and defines a novel indicator for adaptive sampling by incorporating the trait of committor functions.

## 3 NEURAL NETWORK SOLVER FOR COMMITTOR FUNCTIONS

The neural network approximation of partial differential equations involves minimizing a proper loss functional, e.g., the residual loss (Sirignano & Spiliopoulos, 2018; Raissi et al., 2019; Karniadakis et al., 2021) or the variational loss (E & Yu, 2018; Liao & Ming, 2021; Lu et al., 2021). For the

committor function, we consider the variational loss (Li et al., 2019) instead of the residual loss. The variational loss involves up to first-order derivatives of the committer function while the residual loss needs to compute the second-order derivatives, in other words, computing the residual loss is more expensive, especially for high dimensional problems (large $d$ in equation 2). Let $q_{\boldsymbol{\theta}}(\boldsymbol{x})$ be a neural network parameterized with $\boldsymbol{\theta}$, where the input of the neural network is the state variable $\boldsymbol{x}$. One can solve the following variational problem to approximate the committor function

$$\min_{\boldsymbol{\theta}} \int_{\Omega \backslash (A \cup B)} |\nabla q_{\boldsymbol{\theta}}(\boldsymbol{x})|^2 e^{-\beta V(\boldsymbol{x})} d\boldsymbol{x},$$
$$\text{s.t. } q_{\boldsymbol{\theta}}(\boldsymbol{x}) = 0, \boldsymbol{x} \in A; q_{\boldsymbol{\theta}}(\boldsymbol{x}) = 1, \boldsymbol{x} \in B. \tag{3}$$

The details of the derivation of equation 3 can be found in Appendix A.1. We then obtain the following unconstrained optimization problem by adding a penalty term

$$\min_{\boldsymbol{\theta}} \int_{\Omega \backslash (A \cup B)} |\nabla q_{\boldsymbol{\theta}}(\boldsymbol{x})|^2 e^{-\beta V(\boldsymbol{x})} d\boldsymbol{x} + \lambda \left( \int_A q_{\boldsymbol{\theta}}(\boldsymbol{x})^2 p_A(\boldsymbol{x}) d\boldsymbol{x} + \int_B (1 - q_{\boldsymbol{\theta}}(\boldsymbol{x}))^2 p_B(\boldsymbol{x}) d\boldsymbol{x} \right), \tag{4}$$

where $\lambda > 0$ is a penalty parameter, $p_A$ and $p_B$ are two probability density functions on $A$ and $B$ respectively.

To optimize the above variational problem, one needs to generate some random collocation points from a proper probability distribution to estimate the integral in equation 3. One choice is to sample collocation points from the Gibbs measure $e^{-\beta V(\boldsymbol{x})}/Z$, where $Z = \int_{\Omega \backslash (A \cup B)} e^{-\beta V(\boldsymbol{x})} d\boldsymbol{x}$ is the normalization constant, and this can be done by simulating the SDE defined in equation 1. However, generating collocation points by the SDE is inefficient for approximating the committor function, especially for chemical systems with low temperatures (or high energy barriers). This is because the committor function focuses on the transition area while the samples generated by the Langevin dynamics (equation 1) cluster around the metastable regions $A$ and $B$. This implies that the samples from the SDE may not include sufficient effective samples for training $q_{\boldsymbol{\theta}}$. Hence, we need a strategy to seek more effective samples to approximate the committor function, which will be presented in the next section.

Now suppose that we have a set of collocation points $\mathsf{S} = \{\boldsymbol{x}_i\}_{i=1}^N$, where each $\boldsymbol{x}_i \in \Omega \backslash (A \cup B)$ is drawn from a certain probability distribution $p$, and two sets of collocation points $\mathsf{S}_A = \{\boldsymbol{x}_{A,i}\}_{i=1}^{N_A}$ and $\mathsf{S}_B = \{\boldsymbol{x}_{B,i}\}_{i=1}^{N_B}$, where each $\boldsymbol{x}_{A,i}$ and each $\boldsymbol{x}_{B,i}$ are drawn from $p_A$ and $p_B$ respectively. The optimization problem 4 can be discretized as follows

$$\min_{\boldsymbol{\theta}} \frac{1}{N} \sum_{i=1}^N |\nabla q_{\boldsymbol{\theta}}(\boldsymbol{x}_i)|^2 \frac{e^{-\beta V(\boldsymbol{x}_i)}}{p(\boldsymbol{x}_i)} + \lambda \left( \frac{1}{N_A} \sum_{i=1}^{N_A} q_{\boldsymbol{\theta}}(\boldsymbol{x}_{A,i})^2 + \frac{1}{N_B} \sum_{i=1}^{N_B} (q_{\boldsymbol{\theta}}(\boldsymbol{x}_{B,i}) - 1)^2 \right). \tag{5}$$

The key point here is to choose an effective set $\mathsf{S}$ to train $q_{\boldsymbol{\theta}}$. In the next section, we will show how to adaptively generate effective collocation points (a high-quality dataset) on rare transition paths, based on which we expect to improve the accuracy of the approximate solution of equation 2. Considering that the main difficulties come from the transition state region, we will focus on how to choose $\mathsf{S}$ and assume that the integral on the boundary is well approximated by two prescribed sets $\mathsf{S}_A$ and $\mathsf{S}_B$. For simplicity, we will ignore the penalty term when discussing our method.

## 4 DEEP ADAPTIVE SAMPLING ON TRANSITION PATHS

Our goal is to adaptively generate more effective data points distributed in the region of the transition state. This is achieved by designing a deep adaptive sampling method on the transition path.

**Main Idea.** Suppose that at $k$-th step, we have obtained the current approximate solution $q_{\boldsymbol{\theta}_k}$ with $\mathsf{S}_k$. We want to use the information of $q_{\boldsymbol{\theta}_k}$ and the potential function $V$ to detect where the transition area is, based on which we expect to generate some data points in the transition state region that can effectively improve the discretization given by $\mathsf{S}_k$. We then refine $\mathsf{S}_k$ to get $\mathsf{S}_{k+1}$ for the next training step. The more effective data points in the transition area we have, the more accurate solution $q_{\boldsymbol{\theta}}$ we can obtain. To achieve this, we define a proper probability distribution for sample generation based on the following observations: First, $|\nabla_{\boldsymbol{x}} q|^2$ has a peak in the transition state region, implying that

more data points should be introduced around the peak. Second, we may lower the energy barrier to facilitate transitions between the metastable states, which can be done by adding a biased potential $V_{\text{bias}}$ to the original potential $V$ (Li et al., 2019; Kang et al., 2024).

**Sample Generation.** Let $p_{V,q}$ be a probability density function (PDF) that is dependent on $V$ and $q_{\boldsymbol{\theta}}$. Here, we give two choices for constructing $p_{V,q}$. One choice is to set

$$p_{V,q}(\boldsymbol{x}) = \frac{|\nabla q_{\boldsymbol{\theta}}(\boldsymbol{x})|^2 e^{-\beta V(\boldsymbol{x})}}{C_1}, \qquad (6)$$

where $C_1$ is the normalization constant. That is, we treat the nonnegative integrand in equation 3 as an unnormalized probability density function. If there exists a high energy barrier, we can use a biased potential $V_{\text{bias}}$ to lower the energy barrier, which yields the following sampling distribution

$$p_{V,q}(\boldsymbol{x}) = \frac{|\nabla_{\boldsymbol{x}} q_{\boldsymbol{\theta}}(\boldsymbol{x})|^2 e^{-\beta(V(\boldsymbol{x})+V_{\text{bias}}(\boldsymbol{x}))}}{C_2}, \qquad (7)$$

where $C_2$ is the corresponding normalization constant. The biased potential can be chosen to be an umbrella potential (Kästner, 2011) or a potential derived from the metadynamics (Bussi & Laio, 2020; Barducci et al., 2008). The above two sampling distributions can be applied to collective variables (the dimensionality of collective variables is smaller than that of $\boldsymbol{x}$) to achieve the dimension reduction, which will reduce the computational complexity. Suppose that there exist some collective variables $S(\boldsymbol{x}) = [s_1(\boldsymbol{x}), \ldots, s_m(\boldsymbol{x})]^\top$ with $m \ll d$. We can restrict our attention to the collective variables in equation 6 and equation 7, i.e., $p_{V,q}(\boldsymbol{x}) = p_{V,q}(S(\boldsymbol{x}))$. For interested readers, we refer to (Fiorin et al., 2013) for more details of this method. The collective variable method will be applied to the numerical study in section 5.3.

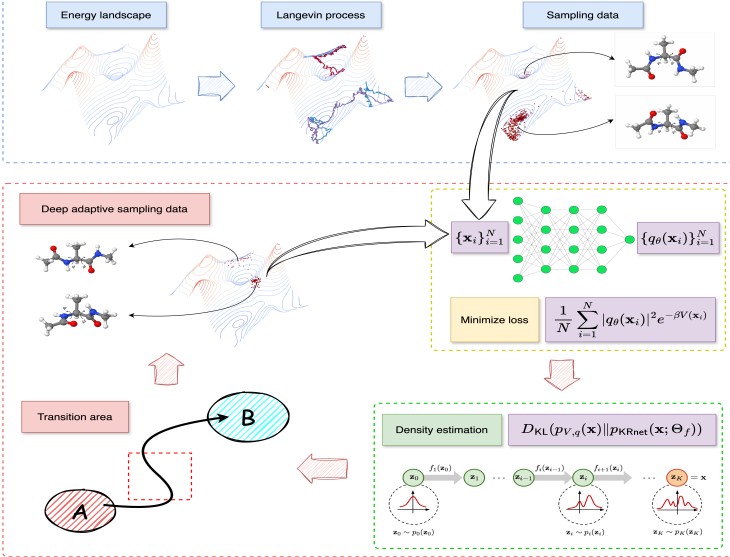

Figure 1: **The schematic of DASTR for computing the committor function.** Training a deep neural network $q_{\boldsymbol{\theta}}$ to approximate the high-dimensional committor function must use a high-quality dataset (i.e. data points from the transition area). The key point is to define a sampling distribution $p_{V,q}$ dependent on the current approximate solution and the potential. Effective data points in the transition area are generated by sampling from $p_{V,q}$, which is achieved through training a deep generative model.

Now the question is how can we generate samples from the above sampling distribution? Here, we use KRnet, which is a type of flow-based generative models (Dinh et al., 2016; Kingma & Dhariwal, 2018), for PDF approximation and sample generation. We note that other deep generative models with exact likelihood computation (Chen et al., 2018; Song et al., 2021) can also be used here. Let $p_{\text{KRnet}}(\boldsymbol{x}; \Theta_f)$ be a PDF model induced by KRnet with parameters $\Theta_f$ (Tang et al., 2020; Wan & Wei, 2022; Tang et al., 2022; 2023). The PDF model $p_{\text{KRnet}}$ is induced by a bijection $f_{\text{KRnet}}$ with parameters $\Theta_f$:

$$p_{\text{KRnet}}(\boldsymbol{x}; \Theta_f) = p_{\boldsymbol{Z}}(f_{\text{KRnet}}(\boldsymbol{x})) \left| \det \nabla_{\boldsymbol{x}} f_{\text{KRnet}} \right|,$$

where $p_{\boldsymbol{Z}}$ is a prior PDF (e.g., the standard Gaussian distribution). We can approximate $p_{V,q}$ through solving the optimization problem

$$\Theta_f^* = \arg \min_{\Theta_f} D_{\text{KL}}(p_{V,q}(\boldsymbol{x}) \| p_{\text{KRnet}}(\boldsymbol{x}; \Theta_f)),$$

where $D_{\text{KL}}(\cdot \| \cdot)$ indicates the Kullback-Leibler (KL) divergence between two distributions. Minimizing the KL divergence is equivalent to minimizing the cross entropy between $p_{V,q}$

and $p_{\mathsf{KRnet}}$ (De Boer et al., 2005; Rubinstein & Kroese, 2013): $H(p_{V,q}, p_{\mathsf{KRnet}}) = -\int_{\Omega \backslash (A \cup B)} p_{V,q}(\boldsymbol{x}) \log p_{\mathsf{KRnet}}(\boldsymbol{x}; \Theta_f) d\boldsymbol{x}$. Since the samples from $p_{V,q}$ are not available, one can approximate the cross entropy using the importance sampling technique:

$$H(p_{V,q}, p_{\mathsf{KRnet}}) \approx -\frac{1}{N} \sum_{i=1}^{N} \frac{p_{V,q}(\boldsymbol{x}_i)}{p_{\mathsf{KRnet}}(\boldsymbol{x}_i; \Theta_f')} \log p_{\mathsf{KRnet}}(\boldsymbol{x}_i; \Theta_f), \tag{8}$$

where $p_{\mathsf{KRnet}}(\boldsymbol{x}_i; \Theta_f')$ is a known PDF model to efficiently generate the samples $\{\boldsymbol{x}_i\}_{i=1}^{N}$ by the KRnet, i.e.,

$$\boldsymbol{x}_i = f_{\mathsf{KRnet}}^{-1}(\boldsymbol{z}_i), \tag{9}$$

with $\boldsymbol{z}_i$ being sampled from the standard Gaussian distribution. We then minimize the discretized cross entropy equation 8 to obtain an approximation of $\Theta_f^*$.

**DASTR Algorithm.** Our strategy is stated as follows. Let $\mathsf{S}_0 = \{\boldsymbol{x}_{0,i}\}_{i=1}^{N_0}$ be a set of collocation points that are sampled from a given distribution $p_0(\boldsymbol{x})$ (say the Gibbs distribution) in $\Omega \backslash (A \cup B)$. Using $\mathsf{S}_0$, we minimize the empirical loss defined in equation 5 to obtain $q_{\boldsymbol{\theta}_1}$. With $q_{\boldsymbol{\theta}_1}$, we minimize the cross entropy in equation 8 to get $p_1 = p_{\mathsf{KRnet}}(\boldsymbol{x}; \Theta_f^{*,(1)})$. A new set $\mathsf{S}_1^g = \{\boldsymbol{x}_{1,i}\}_{i=1}^{n_1}$ with $n_1 \leq N_0$ is generated by $f_{\mathsf{KRnet}}^{-1}(\boldsymbol{z}_i; \Theta_f^{*,(1)})$ (see equation 9) to refine the training set. To be more precise, we replace $n_1$ points in $\mathsf{S}_0$ with $\mathsf{S}_1^g$ to get a new set $\mathsf{S}_1$. Then we continue to update the approximate solution $q_{\boldsymbol{\theta}_1}$ using $\mathsf{S}_1$ as the training set.

In general, at the $k$-stage, suppose that we have $n_j$ samples $\mathsf{S}_j^g = \{\boldsymbol{x}_{j,i}\}_{i=1}^{n_j}$ from $p_j$ for $j = 1, \ldots, k$, where $p_j$ is the PDF model at the $j$-th stage and it can be trained by letting $p_{j-1} = p_{\mathsf{KRnet}}(\boldsymbol{x}_i; \Theta_f')$ in equation 8. The training set $\mathsf{S}_k$ at the $k$-th stage consists of $\boldsymbol{x}_{j,i}$. We use $\mathsf{S}_k$ to obtain $q_{\boldsymbol{\theta}_{k+1}}$ as

$$\boldsymbol{\theta}_{k+1} = \arg\min_{\boldsymbol{\theta}} \sum_{j=0}^{k} \frac{1}{n_j} \sum_{i=1}^{n_j} \alpha_j |\nabla q_{\boldsymbol{\theta}}(\boldsymbol{x}_{j,i})|^2 \frac{e^{-\beta V(\boldsymbol{x}_{j,i})}}{p_j(\boldsymbol{x}_{j,i})} \tag{10}$$

where $q_{\boldsymbol{\theta}}$ is initialized as $q_{\boldsymbol{\theta}_k}$, $\alpha_j = n_j / \sum_{j=0}^{k} n_j$ is a weight to balance the different distributions $p_j$, and $n_0$ is the number of points kept in $\mathsf{S}_0$ at the $k$-th stage. Starting with $p_k = p_{\mathsf{KRnet}}(\boldsymbol{x}; \Theta_f^{*,(k)})$, the density model $p_{\mathsf{KRnet}}(\boldsymbol{x}; \Theta_f)$ is updated by equation 8 to get $p_{k+1}$. A new set $\mathsf{S}_{k+1}^g = \{\boldsymbol{x}_{k+1,i}\}_{i=1}^{n_{k+1}}$ of collocation points is generated by equation 9. We then use $\mathsf{S}_{k+1}^g$ to refine the training set to get $\mathsf{S}_{k+1}$. We repeat the above procedure to obtain Algorithm 1 for the deep adaptive sampling on transition paths. We call this method DASTR for short. The main idea of our algorithm is also illustrated in Figure 1.

---

**Algorithm 1** DASTR

**Input:** Initial $q_{\boldsymbol{\theta}_0}$, maximum stage number $N_{\text{adaptive}}$, maximum epoch number $N_e$, batch size $m$, initial training set $\mathsf{S}_0 = \{\boldsymbol{x}_{0,i}\}_{i=1}^{N_0}$.
  **for** $k = 0 : N_{\text{adaptive}} - 1$ **do**
    **for** $i = 1 : N_e$ **do**
      **for** $l$ steps **do**
        Sample $m$ samples from $\mathsf{S}_k$.
        Update $q_{\boldsymbol{\theta}}(\boldsymbol{x})$ by descending the stochastic gradient of the discrete variational loss (see equation 10).
      **end for**
    **end for**
    **for** $i = 1 : N_e$ **do**
      **for** $l$ steps **do**
        Sample $m$ samples from $\mathsf{S}_k$.
        Update $p_{\mathsf{KRnet}}(\boldsymbol{x}; \Theta_f)$ by descending the stochastic gradient of $H(p_{V,q}, p_{\mathsf{KRnet}})$ (see equation 8).
      **end for**
    **end for**
    Refine the training set: use $p_{k+1}$ to get $\mathsf{S}_{k+1}$.
  **end for**
**Output:** $q_{\boldsymbol{\theta}}$

---

## 5 NUMERICAL STUDY

We conduct three numerical experiments to demonstrate the effectiveness of the proposed method. The first one is a 10-dimensional rugged Mueller potential problem, the second one is a 20-dimensional standard Brownian motion problem, and the last one is the alanine dipeptide problem with the dimension $d = 66$. The detailed settings of numerical experiments are provided in Appendix A.2.

## 5.1 RUGGED MUELLER POTENTIAL

We consider the extended rugged Mueller potential embedded in the 10-dimensional space, which is a well-known test problem in computational chemical physics (Li et al., 2019; 2022). The extended rugged Mueller potential is given by $V(\boldsymbol{x}) = V_{\mathrm{rm}}(x_1, x_2) + 1/(2\sigma^2)\sum_{i=3}^{10} x_i^2$, where $\boldsymbol{x} = [x_1, x_2, \ldots, x_{10}]$ and $V_{\mathrm{rm}}(x_1, x_2)$ is the rugged Mueller potential defined in $[-1.5, 1] \times [-0.5, 2]$

$$V_{\mathrm{rm}}(x_1, x_2) = \sum_{i=1}^{4} D_i e^{a_i(x_1-\xi_i)^2 + b_i(x_1-\xi_i)(x_2-\eta_i) + c_i(x_2-\eta_i)^2} + \gamma\sin(2k\pi x_1)\sin(2k\pi x_2).$$

We set $\sigma = 0.05$ as in (Li et al., 2019), and the other parameters are set to be the same as in (Lai & Lu, 2018). The inverse temperature is set to $\beta = 1/10$. In this test problem, the two metastable sets $A$ and $B$ are two cylinders with centers $[x_1, x_2] = [-0.558, 1.441]$ and $[x_1, x_2] = [0.623, 0.028]$ respectively and radius $0.1$. In this setting, the solution of this 10-dimensional problem is the same as that of the two-dimensional rugged Mueller potential, i.e., $q(\boldsymbol{x}) = q_{\mathrm{rm}}(\boldsymbol{x})$ (Li et al., 2019; 2022). So, we can use the finite element method implemented in FEniCS (Alnæs et al., 2015; Logg et al., 2012) to obtain a reference solution to evaluate the performance. For comparison, we also implement the artificial temperature method (Li et al., 2019) as the baseline model. Here we define the $L^2$ relative error $\|\boldsymbol{q_\theta} - \boldsymbol{q}_{\mathrm{ref}}\|_2 / \|\boldsymbol{q}_{\mathrm{ref}}\|_2$, where $\boldsymbol{q_\theta}$ and $\boldsymbol{q}$ denote two vectors whose elements are the function values of $q_\theta$ and $q_{\mathrm{ref}}$ at some grids respectively. The settings of neural networks and training details can be found in Appendix A.2.1.

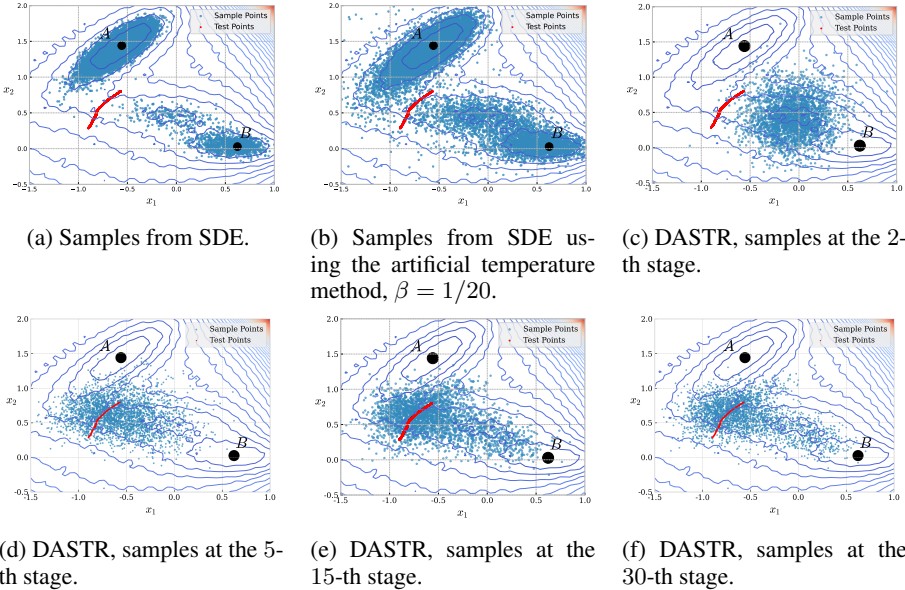

(a) Samples from SDE.

(b) Samples from SDE using the artificial temperature method, $\beta = 1/20$.

(c) DASTR, samples at the 2-th stage.

(d) DASTR, samples at the 5-th stage.

(e) DASTR, samples at the 15-th stage.

(f) DASTR, samples at the 30-th stage.

Figure 2: DASTR, samples for the 10-dimensional rugged Mueller potential problem. The red line denotes the test points from the $1/2$-isosurface ($q \approx 1/2$) projected onto the $x_1$-$x_2$ plane.

Figure 2 shows the samples from different sampling strategies, where these samples are projected onto the $x_1$-$x_2$ plane. Specifically, Figure 2a shows the samples generated by SDE defined in equation 1. It can be seen that the samples from SDE are located around the two metastable states $A$ and $B$, which is not able to provide effective samples to approximate the committor function. Figure 2b shows the samples from SDE with the artificial temperature method. While more samples show up in the transition state region compared with Figure 2a, there still does not exist sufficient information in the dataset to capture the committor function well. Our method is able to provide effective samples in the transition area. As shown in Figures 2c-2f, the evolution of the training set with respect to adaptivity iterations $k = 2, 5, 15, 30$ is presented, where we randomly select 5000 samples in the training set for visualization. Obviously, such samples are distributed in the transition state region ($\Omega\backslash(A \cup B)$), which is desired for computing the committor function.

Figure 3a shows the error behavior of different methods. In Figure 3b-3d, we compare the reference solution $q_{\mathrm{ref}}$ obtained by the finite element method, the DASTR solution given by $4 \times 10^5$ samples

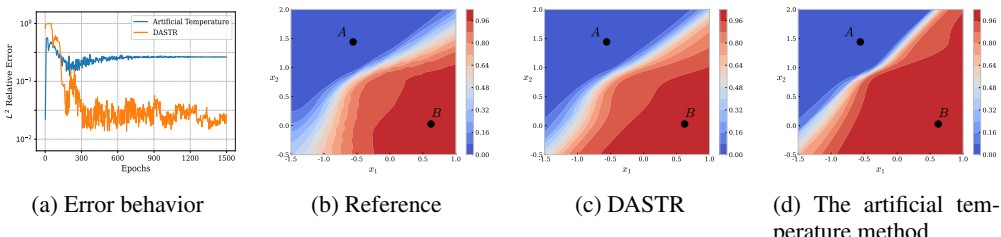

(a) Error behavior     (b) Reference     (c) DASTR     (d) The artificial temperature method

Figure 3: Solutions, 10-dimensional rugged Mueller potential test problem.

and the approximate solution given by $4 \times 10^5$ samples with the artificial temperature method. Figure 4 shows the relative errors with respect to different sample sizes. From Figure 4, it is seen that the DASTR method is much more accurate than the method of sampling from dynamics. Due to the difficulty of sampling in the transition state region using SDE with the artificial temperature method, the solution obtained through the artificial temperature method fails to accurately capture the information of the committor function in the transition state region. To further investigate the performance of the proposed method, in Table 1, we show the $L^2$ relative errors of neural networks with varying numbers of neurons subject to different sample sizes. Here, we sample 12099 points near the $1/2$-isosurface ( $q(\boldsymbol{x}) \approx 0.5$ ) to compute the relative error. Our DASTR method is one order of magnitude more accurate than the baseline method in all settings.

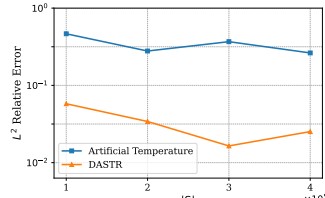

Figure 4: The error w.r.t. sample size $|\mathsf{S}|$.

Table 1: 10-dimensional rugged Mueller potential test problem: errors for different settings of neural networks and sampling strategies. We take 4 independent runs to compute the error statistics (mean $\pm$ standard deviation).

| Sampling Method | $|\mathsf{S}|$ | Number of Neurons in Hidden Layer | | |
| --- | --- | --- | --- | --- |
| | | 20 | 50 | 100 |
| SDE with the artificial temperature method | $1 \times 10^5$ | $0.5446 \pm 0.0724$ | $0.4693 \pm 0.0627$ | $0.4023 \pm 0.0819$ |
| | $2 \times 10^5$ | $0.3183 \pm 0.0592$ | $0.2677 \pm 0.0708$ | $0.3063 \pm 0.0477$ |
| | $3 \times 10^5$ | $0.2717 \pm 0.0487$ | $0.2780 \pm 0.0584$ | $0.3955 \pm 0.0311$ |
| | $4 \times 10^5$ | $0.3822 \pm 0.0555$ | $0.3019 \pm 0.0649$ | $0.3822 \pm 0.1213$ |
| DASTR (this work) | $1 \times 10^5$ | $0.0620 \pm 0.0070$ | $0.0602 \pm 0.0113$ | $0.0615 \pm 0.0071$ |
| | $2 \times 10^5$ | $0.0498 \pm 0.0102$ | $0.0443 \pm 0.0049$ | $0.0310 \pm 0.0024$ |
| | $3 \times 10^5$ | $0.0386 \pm 0.0089$ | $0.0412 \pm 0.0091$ | $0.0172 \pm 0.0028$ |
| | $4 \times 10^5$ | $0.0371 \pm 0.0056$ | $0.0343 \pm 0.0065$ | $0.0206 \pm 0.0052$ |

## 5.2 STANDARD BROWNIAN MOTION

In this test problem, we consider the committor function under the standard Brownian motion (Hartmann et al., 2019; Nüsken & Richter, 2023). For a stochastic process $(\boldsymbol{X}_t)_{t \geq 0} \in \mathbb{R}^d$, which is a standard Brownian motion starting at $\boldsymbol{x} \in \mathbb{R}^d$, that is, $\boldsymbol{X}_t = \boldsymbol{x} + \boldsymbol{W}_t$, corresponding to $\nabla V(\boldsymbol{X}_t) = 0$ and $\beta = 1/2$ in equation 1. The two metastable sets $A$ and $B$ are defined as $A = \{\boldsymbol{x} \in \mathbb{R}^d : \|\boldsymbol{x}\|_2 < a\}, B = \{\boldsymbol{x} \in \mathbb{R}^d : \|\boldsymbol{x}\|_2 > b\}$ with $b > a > 0$. With these settings, for $d \geq 3$, there exists an analytical solution $q(\boldsymbol{x}) = (a^2 - \|\boldsymbol{x}\|_2^{2-d} a^2)/(a^2 - b^{2-d} a^2)$. In this test problem, we set $d = 20$ and $a = 1, b = 2$. The settings of neural networks and training details can be found in Appendix A.2.2. Since the solution to this test problem cannot be projected onto the low-dimensional space, we here compare different sampling methods by computing the $L^2$ relative error at a validation set with 5000 data points along a curve $\{(\kappa, \dots, \kappa)^\top : \kappa \in [a/\sqrt{d}, b/\sqrt{d}]\}$ (Nüsken & Richter, 2023).

Figure 5 shows the results of the 20-dimensional standard Brownian motion test problem. Specifically, Figure 5a shows the solutions obtained by different sampling methods, where it can be seen

that the DASTR solution is more accurate than those of other sampling strategies. Figure 5b shows the behavior of relative errors during training, where DASTR performs better than the uniform sampling strategy and SDE. Figure 5c shows the relative errors for the uniform sampling method, SDE, and DASTR, where different numbers of samples are tested. From Figure 5c, it is clear that, as the number of samples increases, the relative error of DASTR decreases more quickly than those of SDE and the uniform sampling strategy.

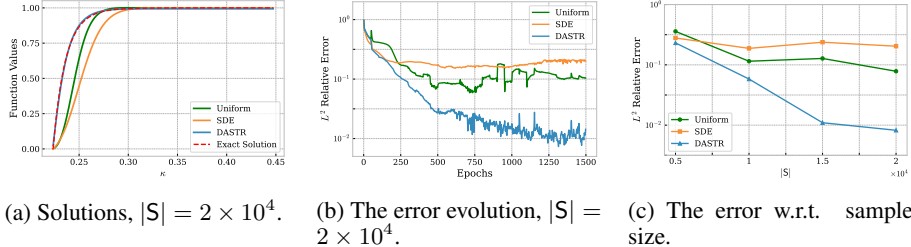

(a) Solutions, $|S| = 2 \times 10^4$.

(b) The error evolution, $|S| = 2 \times 10^4$.

(c) The error w.r.t. sample size.

Figure 5: Solutions evaluated along a curve and the behavior of relative errors, 20-dimensional standard Brownian motion test problem. The relative error is computed at the points along the curve $\{(\kappa, \ldots, \kappa)^\top : \kappa \in [a/\sqrt{d}, b/\sqrt{d}]\}$.

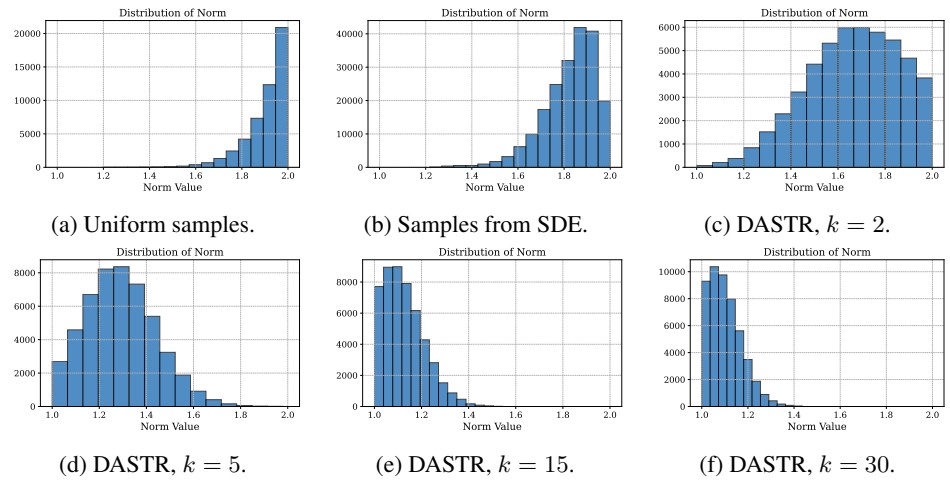

(a) Uniform samples.

(b) Samples from SDE.

(c) DASTR, $k = 2$.

(d) DASTR, $k = 5$.

(e) DASTR, $k = 15$.

(f) DASTR, $k = 30$.

Figure 6: Histogram of the norm of samples, 20-dimensional test problem.

To see why DASTR works well, let us visualize the $L^2$-norm of samples from different sampling strategies. Figure 6 shows the histogram of the norm of samples for different sampling strategies. From Figure 6a and Figure 6b, we can see that most of the samples fall into the interval where the norm of samples is near 2. This means that it is difficult to generate samples in the transition state region using the uniform sampling strategy or SDE. Indeed, in high-dimensional spaces, most of the volume of an object concentrates around its surface (Vershynin, 2018; Wright & Ma, 2022). Hence, using uniform samples or samples generated by SDE is inefficient for estimating the committor function. Figures 6c, 6d, 6e, and 6f show the histogram of the norm of samples from DASTR. These histograms imply that the samples from DASTR capture the information of transitions, which improves the accuracy of estimating the committor function. In Table 2, we again present the $L^2$ relative errors of neural networks with varying numbers of neurons subject to different sample sizes. Our DASTR method is one order of magnitude more accurate than the baseline methods in most settings.

### 5.3 ALANINE DIPEPTIDE

In this part, the isomerization process of the alanine dipeptide in vacuum at $T = 300K$ is studied. This test problem is a benchmark in various literatures (Li et al., 2019; Kang et al., 2024).

Table 2: 20-dimensional standard Brownian motion test problem: error for different settings of neural networks and sampling strategies. We take 4 independent runs to compute the statistics of the error (mean ± standard deviation).

| Sampling Method | $|S|$ | Number of Neurons in Hidden Layer | | |
| --- | --- | --- | --- | --- |
| | | 20 | 50 | 100 |
| Uniform | $5 \times 10^3$ | $0.1767 \pm 0.0240$ | $0.1906 \pm 0.0214$ | $0.4555 \pm 0.0557$ |
| | $1 \times 10^4$ | $0.1861 \pm 0.0319$ | $0.1760 \pm 0.0492$ | $0.1310 \pm 0.0197$ |
| | $1.5 \times 10^4$ | $0.2125 \pm 0.0220$ | $0.2003 \pm 0.0295$ | $0.1454 \pm 0.0609$ |
| | $2 \times 10^4$ | $0.1963 \pm 0.0866$ | $0.1611 \pm 0.0227$ | $0.1402 \pm 0.0515$ |
| SDE | $5 \times 10^3$ | $0.2127 \pm 0.0802$ | $0.2641 \pm 0.0416$ | $0.3696 \pm 0.0633$ |
| | $1 \times 10^4$ | $0.2846 \pm 0.0523$ | $0.2606 \pm 0.0343$ | $0.1586 \pm 0.0179$ |
| | $1.5 \times 10^4$ | $0.2861 \pm 0.0177$ | $0.1865 \pm 0.0220$ | $0.1706 \pm 0.0434$ |
| | $2 \times 10^4$ | $0.2321 \pm 0.0278$ | $0.1864 \pm 0.0254$ | $0.1342 \pm 0.0434$ |
| DASTR (this work) | $5 \times 10^3$ | $0.0996 \pm 0.0374$ | $0.1073 \pm 0.0128$ | $0.0266 \pm 0.1396$ |
| | $1 \times 10^4$ | $0.0835 \pm 0.0215$ | $0.0415 \pm 0.0167$ | $0.0410 \pm 0.0106$ |
| | $1.5 \times 10^4$ | $0.0824 \pm 0.0412$ | $0.0197 \pm 0.0045$ | $0.0141 \pm 0.0053$ |
| | $2 \times 10^4$ | $0.0227 \pm 0.0051$ | $0.0209 \pm 0.0096$ | $0.0114 \pm 0.0021$ |

The molecule we consider here consists of 22 atoms, each of which has three coordinates. This means that the dimension of the state variable is $d = 66$ in equation 2. There are two important dihedrals related to their configurations: $\phi$ (C-N-CA-C) and $\psi$ (N-CA-C-N), i.e., the collective variables. The two metastable conformers of the molecule are $C_{7eq}$ and $C_{ax}$ located around $(\phi, \psi) = (-85°, 75°)$ and $(72°, -75°)$ respectively. More specifically, the two metastable sets $A$ and $B$ are defined as (Li et al., 2019): $A = \{\boldsymbol{x} : \|(\phi(\boldsymbol{x}), \psi(\boldsymbol{x})) - C_{7eq}\|_2 < 10°\}$, $B = \{\boldsymbol{x} : \|(\phi(\boldsymbol{x}), \psi(\boldsymbol{x})) - C_{ax}\|_2 < 10°\}$. In Figure 7, the molecule structures of two metastable states and two transition states are displayed.

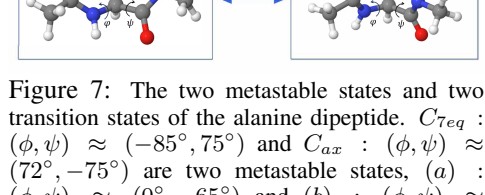

Figure 7: The two metastable states and two transition states of the alanine dipeptide. $C_{7eq}$ : $(\phi, \psi) \approx (-85°, 75°)$ and $C_{ax}$ : $(\phi, \psi) \approx (72°, -75°)$ are two metastable states, $(a)$ : $(\phi, \psi) \approx (0°, -65°)$ and $(b)$ : $(\phi, \psi) \approx (130°, -125°)$ are two transition states.

The goal is to compute the committor function under the CHARMM force filed (Jo et al., 2008; Brooks et al., 2009; Lee et al., 2016). Due to the high energy barrier between the two metastable states $A$ and $B$, it is almost impossible for the molecule to cross this barrier from $A$ to $B$. Consequently, sampling in the transition state region is extremely challenging. Nevertheless, generating samples in the transition area is crucial for training neural networks to approximate the committor function. Moreover, for this realistic problem, we need to ensure that the samples from deep generative models obey the molecular configuration, which makes this problem much more challenging to solve. To handle such a tricky situation, we combine our DASTR method with the umbrella sampling method (Kästner, 2011) and the collective variable method. Simply speaking, we use the proposed DASTR method to generate the target collective variables used in the umbrella potential. The details of the overall procedure can be found in Appendix A.2.3.

For this problem, it is intractable to obtain the reference solution with grid-based numerical methods. To assess the performance of our method, we again consider those samples from the $1/2$-isosurface. More specifically, we first use umbrella sampling (see Appendix A.3) to sample $1 \times 10^7$ points. After that, we use the trained model to compute $q_{\boldsymbol{\theta}}$ at these sample points and filter to keep points on the set $\Gamma := \{\boldsymbol{x} : |q_{\boldsymbol{\theta}}(\boldsymbol{x}) - 0.5|\} \leq 5 \times 10^{-5}$. We then select 300 points in $\Gamma$ and conduct 200 simulations of SDE for each point to obtain the corresponding trajectory. By counting the number of times of these points first reaching $B$ before $A$, we can estimate $q$ for such points by the definition of committor functions. If the trained model $q_{\boldsymbol{\theta}}$ is indeed a good approximation of the committor function, then the probability distribution (in fact, we use the relative frequency to represent the true

probability) of reaching $B$ before $A$ should be close to a normal distribution with mean $0.5$ (Chen et al., 2023).

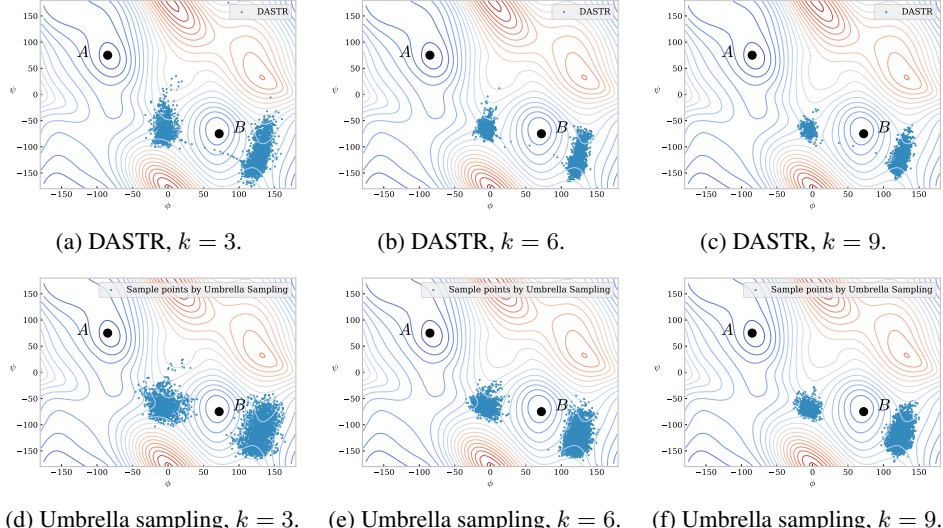

(a) DASTR, $k = 3$.   (b) DASTR, $k = 6$.   (c) DASTR, $k = 9$.

(d) Umbrella sampling, $k = 3$. (e) Umbrella sampling, $k = 6$. (f) Umbrella sampling, $k = 9$.

Figure 8: Samples during training, the alanine dipeptide test problem. We use DASTR to generate target CVs in the transition state region; the umbrella sampling method is employed to generate samples around the target CVs to refine the training set.

The results are shown in Figure 8 and Figure 9. In Figure 8a-8c, we show the candidate samples generated by DASTR. It is clear that these samples are located in the transition state region. To ensure that the samples obey the molecular configuration, we use the umbrella sampling method to refine them as shown in Figure 8d-8f. From Figure 9, it is seen that the approximate committor function values cluster around $1/2$, which indicates that our DASTR method provides a good approximation on the $1/2$-isosurface.

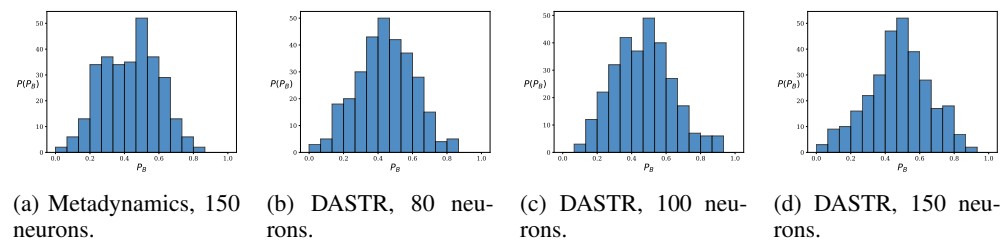

(a) Metadynamics, 150 neurons. (b) DASTR, 80 neurons. (c) DASTR, 100 neurons. (d) DASTR, 150 neurons.

Figure 9: The alanine dipeptide test problem: the histograms of the committor function values on the $1/2$-th isosurface of $q_\theta$ with different numbers of neurons. $q_\theta$ is a five-layer fully connected neural network. The training details can be found in Appendix A.2.3.

# 6  CONCLUSION

We developed a novel deep adaptive sampling approach on rare transition paths (DASTR) for estimating the high-dimensional committor function. With DASTR, the scarcity of effective data points can be addressed, and the performance of neural network approximation for the high-dimensional committor function is improved significantly.

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

## A  APPENDIX

### A.1  DERIVATION OF VARIATIONAL FORMULATION

Let $u = q + \gamma\eta$ be the result of a perturbation $\gamma\eta$ of $q$, where $\gamma$ is small and $\eta$ is a differentiable function. Since $q$ is the minimizer of equation 3, for any $\eta$, we have

$$
\begin{aligned}
0 &= \frac{1}{2}\frac{\partial}{\partial\gamma}|_{\gamma=0}\int_{\Omega\backslash(A\cup B)}|\nabla u(\boldsymbol{x})|^2 e^{-\beta V(\boldsymbol{x})}d\boldsymbol{x} \\
&= \int_{\Omega\backslash(A\cup B)}\nabla q(\boldsymbol{x})\cdot\nabla\eta(\boldsymbol{x})e^{-\beta V(\boldsymbol{x})}d\boldsymbol{x} \\
&= \int_{\Omega\backslash(A\cup B)}\nabla\cdot\left(\nabla q(\boldsymbol{x})\eta(\boldsymbol{x})e^{-\beta V(\boldsymbol{x})}\right)d\boldsymbol{x} - \int_{\Omega\backslash(A\cup B)}\eta(\boldsymbol{x})\nabla\cdot\left(\nabla q(\boldsymbol{x})e^{-\beta V(\boldsymbol{x})}\right)d\boldsymbol{x} \quad (11) \\
&= -\int_{\Omega\backslash(A\cup B)}\eta(\boldsymbol{x})\nabla\cdot\left(\nabla q(\boldsymbol{x})e^{-\beta V(\boldsymbol{x})}\right)d\boldsymbol{x} \\
&= -\int_{\Omega\backslash(A\cup B)}\eta(\boldsymbol{x})e^{-\beta V(\boldsymbol{x})}\left(\Delta q(\boldsymbol{x}) - \beta\nabla V(\boldsymbol{x})\cdot\nabla q(\boldsymbol{x})\right)d\boldsymbol{x},
\end{aligned}
$$

where the fourth equality follows from the integration by parts and the Neumann condition in equation 2. Because equation 11 holds for any $\eta$, we have $\Delta q(\boldsymbol{x}) - \beta\nabla V(\boldsymbol{x})\cdot\nabla q(\boldsymbol{x}) = 0$, which is the desired PDE form of the committor function.

### A.2  IMPLEMENTATION DETAILS

#### A.2.1  RUGGED MUELLER POTENTIAL

We choose a four-layer fully connected neural network $q_{\boldsymbol{\theta}}$ with 100 neurons to approximate the solution, and the activation function of $q_{\boldsymbol{\theta}}$ in hidden layers is set to the hyperbolic tangent function. The activation of the output layer is the sigmoid function. For KRnet, we take five blocks and eight affine coupling layers in each block. A two-layer fully connected neural network with 120 neurons is employed in each affine coupling layer. The activation function of KRnet is the rectified linear unit (ReLU) function. To generate points in $\Omega\backslash(A\cup B)$, we use the KRnet to learn the sampling distribution $p_{V,q}(\boldsymbol{x}) = |\nabla q_{\boldsymbol{\theta}}(\boldsymbol{x})|^2 e^{-\beta V(\boldsymbol{x})}$ in the box $[-1.5, 1]\times[-0.5, 2]\times[-1, 1]^{d-2}$, and then remove points within the region $A$ and $B$. This can be done by adding a logistic transformation layer (Tang et al., 2023) or a new coupling layer proposed in (Zeng et al., 2023). We set $\lambda = 10$ in equation 4. The learning rate for the ADAM optimizer is set to 0.0001, with a decay rate 0.8 applied every 200 epochs during training $q_{\boldsymbol{\theta}}$, and the batch size is set to $m = 5000$. The numbers of adaptivity iterations is set to $N_{\text{adaptive}} = 30$ when $N_e = 50$ in Algorithm 1. In this test problem, we replace all the data points in the current training set with new samples.

It is difficult to sample in the transition region when directly running the simulation of SDE. We implement the artificial temperature method as the baseline. More specifically, we increase the temperature by setting $\beta' = 1/20$ to obtain the modified SDE. This modified Langevin equation is solved by the Euler-Maruyama scheme with the time step $\Delta t = 10^{-5}$. With this setting, the data points are sampled from the trajectory of the modified Langevin equation. In this example, we compare the results obtained from DASTR with those from the artificial temperature method.

#### A.2.2  STANDARD BROWNIAN MOTION

We choose a four-layer fully connected neural network $q_{\boldsymbol{\theta}}$ with 100 neurons to approximate the solution, and the activation function of $q_{\boldsymbol{\theta}}$ is set to the square of the hyperbolic tangent function. For KRnet, we take five blocks and eight affine coupling layers in each block. A two-layer fully connected neural network with 120 neurons is employed in each affine coupling layer. The activation function of KRnet is the rectified linear unit (ReLU) function. The learning rate for the ADAM optimizer is set to 0.001, with a decay rate 0.8 applied every 200 epochs during training $q_{\boldsymbol{\theta}}$. We set the number of adaptivity iterations to $N_{\text{adaptive}} = 30$, with $N_e = 50$ training epochs per stage. The batch size for training $q_{\boldsymbol{\theta}}$ is set to $m = 1000$ and for training the PDF model is set to $m = 5000$. In the first stage, we generate $N_0$ uniform samples from $\Omega\backslash(A\cup B)$ and $N_0/2$ points each from

$\partial A$ and $\partial B$. For the remaining stages, we select $N_0/2$ points from the uniform samples and $N_0/2$ points from the deep generative model. We set $\lambda = 1000$ in equation 4.

We use the deep generative model to approximate $p_{V,q}(\boldsymbol{x}) = |\nabla q_{\boldsymbol{\theta}}(\boldsymbol{x})|^2 e^{-\beta V(\boldsymbol{x})}$, where the probability density function induced by the deep generative model is defined in the box $[-2, 2]^d$. To ensure points in $\Omega \backslash (A \cup B)$, we just remove points within the region $A$ and $B$ generated by the deep generative model. For comparison, we also use the SDE to generate data points to train $q_{\boldsymbol{\theta}}$, where the Euler-Maruyama scheme with the time step $\Delta t = 10^{-6}$ is applied to get the trajectory.

### A.2.3 ALANINE DIPEPTIDE

In this test problem, we choose the dihedrals $\phi$ (with respect to C-N-CA-C), $\psi$ (with respect to N-CA-C-N) as the collective variables (CVs). For this real example, it is not suitable for using the uniform samples as the initial training set, since uniform samples are not effective for solving this high-dimensional ($d = 66$) problem and also do not obey the molecular configuration. We use metadynamics to generate samples as the initial training set.

Metadynamics is an enhanced sampling technique to explore free energy landscapes of complex systems. The idea of metadynamics is to add a history-dependent biased potential to the system to discourage it from revisiting previously sampled states (Bussi & Laio, 2020; Barducci et al., 2008). This is done by periodically depositing Gaussian potentials along the trajectory of the collective variables (CVs). Mathematically, the Gaussian potential can be expressed as:

$$V_{G,t}(\boldsymbol{x}) = \sum_{t'=0,\tau,2\tau,\dots}^{t'<t} w \exp\left( - \sum_{i=1}^{m} \frac{(S_i(\boldsymbol{x}) - S_i(\boldsymbol{x}_{t'}))^2}{2\sigma_i^2} \right), \tag{12}$$

where $w$ is the height of the Gaussian potential, $\sigma$ is the width of the Gaussian potential, $m$ is number of CVs, and $S_i(\boldsymbol{x}_t)$ denotes the collective variables at time $t$. After adding the above Gaussian potential, we generate samples using the modified potential:

$$V_{\text{modified}}(\boldsymbol{x}) = V(\boldsymbol{x}) + V_{G,t}(\boldsymbol{x}),$$

where $V(\boldsymbol{x})$ is the original potential. That is, the biased potential in equation 7 is the Gaussian potential function $V_{G,t}$. During the simulation, the Gaussian potential lowers the energy barrier, allowing the system to explore more configurations of molecules. So, we can generate effective data points as the initial training set by metadynamics for this alanine dipeptide problem.

We simulate the Langevin dynamics with the time step $\Delta t = 0.2\,\text{fs}$ and a damping coefficient $1\,\text{ps}^{-1}$. One term of the Gaussian potential is added every 500 steps, with parameters $w = 1.0\,\text{kJ/mol}$, $\sigma = 0.1\,\text{rad}$. We finally get a total of 10000 terms in equation 12. Samples are selected outside the regions $A$ and $B$, and system configurations are saved to conduct the importance sampling step in equation 10. The simulation is conducted in OpenMM (Eastman et al., 2017), a molecular dynamics simulation toolkit with high-performance implementation. Figure 10 shows the samples from the original dynamics and metadynamics. From this figure, it is clear that using metadynamics to generate initial data points is better since more samples are located in the transition region.

We choose a five-layer fully connected neural network $q_{\boldsymbol{\theta}}$ (with $80, 100, 150$ neurons) to approximate the solution, and the activation function of hidden layers is set to the hyperbolic tangent function. The activation of the output layer is the sigmoid function. Here, we only use the deep generative model to model the sampling distribution in terms of the collective variables $\phi$ and $\psi$. The trained KRnet is used to generate $S(\boldsymbol{x}_0) = [\phi, \psi]^\top$ in equation 13 (see Appendix A.3). For KRnet, we take one block and six affine coupling layers in each block. A two-layer fully connected neural network with $64$ neurons is employed in each affine coupling layer. The activation function of KRnet is the rectified linear unit (ReLU) function. The learning rate for the ADAM optimizer is set to $0.0001$, with a decay factor of $0.5$ applied every $300$ epochs. We set the batch size $m = 5000$ and $N_e = 300$. The numbers of adaptivity iterations is set to $N_{\text{adaptive}} = 10$. We sample $1.5 \times 10^4$ points in $A$ and $B$ respectively to enforce the boundary condition in the training process for all stages. We set $\lambda = 10$ in equation 4.

We employ KRnet to learn the sampling distribution in equation 7. In the first stage, we train the neural network $q_{\boldsymbol{\theta}}$ using $2 \times 10^5$ points sampled by metadynamics. Then we use these points to train the PDF model induced by KRnet with support $[-180°, 180°]^2$, with the bias potential $V_{\text{bias}}$ in

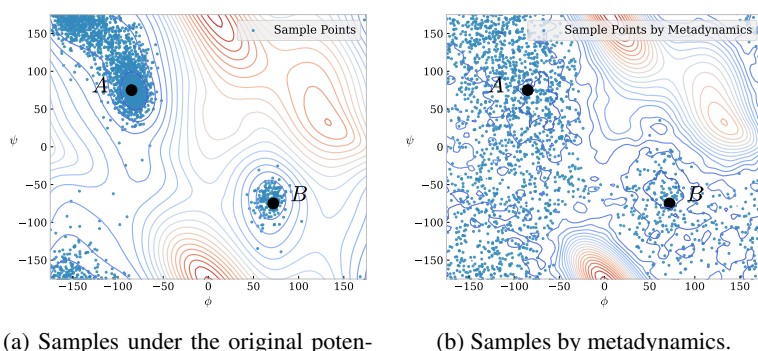

(a) Samples under the original potential.

(b) Samples by metadynamics.

Figure 10: Samples from the original dynamics and metadynamics.

equation 7 being the Gaussian potential $V_{G,t}$ defined in equation 12. In the rest of the stages, we train the neural network $q_{\boldsymbol{\theta}}$ with $5 \times 10^4$ points sampled by umbrella sampling with the bias potential $V_{\mathrm{US}}$ (see Appendix A.3) and $1.5 \times 10^5$ points by metadynamics. We train the KRnet using the same sample points as those of training $q_{\boldsymbol{\theta}}$.

During the training procedure, we increase $k_{\mathrm{us}}$ in equation 13 from 200 kJ/mol to 600 kJ/mol. We sample 100 points for each target CVs in the umbrella sampling procedure. For comparison, we use the solution obtained by training a neural network $q_{\boldsymbol{\theta}}$ with 150 neurons with $2 \times 10^5$ points sampled via metadynamics for 3000 epochs.

### A.3 UMBRELLA SAMPLING

The umbrella sampling method is also an enhanced sampling technique. It introduces external biased potentials to pull the system out of local minima, thereby enabling a more uniform exploration of the entire free energy surface. This method is particularly effective in calculating free energy differences and studying reaction pathways in complex molecular processes. The umbrella sampling method employs a series of biased simulations, dividing the reaction space of collective variables into multiple overlapping windows, where each biased potential is applied in its corresponding window (Kästner, 2011). The umbrella potential is usually defined as:

$$V_{\mathrm{US}}(\boldsymbol{x}) = \frac{1}{2} \sum_{i=1}^{m} k_{\mathrm{us}}(S_i(\boldsymbol{x}) - S_i(\boldsymbol{x}_0))^2, \tag{13}$$

where $S_i(\boldsymbol{x})$ represents the CVs with respect to $\boldsymbol{x}$, $m$ is the number of CVs, and $k_{\mathrm{us}}$ is the force constant. In this work, we focus on sampling in the final window, helping us effectively sample the desired regions of CVs. Therefore, we perform a rapid iterative process of umbrella sampling to transfer the CVs to the target region, and finally sample near the target CVs in the modified potential:

$$V_{\mathrm{modified}}(\boldsymbol{x}) = V(\boldsymbol{x}) + V_{\mathrm{US}}(\boldsymbol{x}),$$

where $V$ is the original potential, and the $S_i(\boldsymbol{x}_0)$ in equation 13 is the target CVs generated by the trained deep generative model.

