# OpenReview forum: "Estimating Committor Functions via Deep Adaptive Sampling on Rare Transition Paths"
_ICLR.cc/2025/Conference — ICLR 2025 Conference Withdrawn Submission_

### Official Review · Reviewer_mCsg · 2024-10-28

**Soundness:** 3
**Presentation:** 3
**Contribution:** 3
**Rating:** 3
**Confidence:** 3

**Summary:**

This paper proposes a methodology, referred to as DASTR, to approximate the committor function, which is central in transition path theory (TPT). The authors adopt a variational loss to approximate the committor function, where the loss is expressed as an integral with an empirical distribution. They aim to increase sampling in the reactive (transition) region by adjusting the sampling measure. Specifically, they adjust the sampling measure to be proportional to the magnitude of the gradient field of the neural-parameterized committor function, focusing on regions with high variation that require denser learning.
They sample from the generative model $p_\text{KRnet}$ and compensate the training objective using the probability density of the generative model for unbiased training. In summary, the training process iteratively learns both the committor function $q_\theta$ and the sampling measure $p_\text{KRnet}$.
The authors evaluate the model’s accuracy (using the $L^2$ norm) by comparing it to numerically obtained committor functions on a 2D rugged Mueller potential (with eight dummy dimensions) and a standard Brownian system (without potential). Additionally, they assess the model on transition paths between the $C_{7eq}$ and $C_{ax}$ basins in alanine dipeptide as a real-world application.

**Strengths:**

- Proposes an efficient framework for approximating the committor function with an adaptive sampling distribution.
- Utilizes a flow-based generative model, which allows exact likelihood computation—a useful approach.

**Weaknesses:**

- Using a generative model as the sampling distribution may not be scalable. Complex reaction systems requiring TPT often involve large molecules, not small systems like those addressed by TST. More investigation is needed to verify the framework’s applicability to larger systems.
- Although the authors describe the system as high-dimensional, the alanine dipeptide system is the smallest among poly-peptides. Moreover, they use collective variables for dimensionality reduction. In many cases, collective variables are not well-known and require extensive chemical analysis of the reaction path. Therefore, it would be beneficial to explore the application of a collective-variable-free model.
- Most evaluations are conducted on the 1/2-isosurface. The committor function could be potentially applied in areas like transition rate analysis or path sampling. To assess the framework’s utility, additional validation metrics such as flux comparison, path sampling quality, and rate comparison should be conducted. One suggestion is to calculate the transition rate of the forward and backward reactions using the committor function and compare it to the equilibrium constant, which could be obtained based on the energetics.
- There is no comparison with other methods. The rugged Mueller potential is widely used in studies approximating the committor function, as referenced by the authors. At minimum, a comparison of training time and accuracy with other methods on this rugged Mueller potential would be valuable.

If the authors address two or more of these weaknesses, I would consider increasing the rating by 3–4 points.

**Questions:**

- What is the reason for evaluating only at the 0.5 isosurface? Wouldn't it be fairer to evaluate at other isosurfaces between 0.0 and 1.0 as well? Since your proposed sampling method focuses more on the 0.5 isosurface, wouldn’t a more refined evaluation be necessary near the A and B basins?

---

> ### Author Response · Authors · 2024-11-15
> **Response to Reviewer mCsg (part 1)**
>
> 1. Using a generative model as the sampling distribution may not be scalable. Complex reaction systems requiring TPT often involve large molecules, not small systems like those addressed by TST. More investigation is needed to verify the framework’s applicability to larger systems.
>
> We agree with this reviewer that the performance of DASTR needs to be further investigated using large reaction systems. Based on our available computational resources, we use some benchmark test problems (e.g. alanine dipeptide), which are studied in existing references, to demonstrate the effectiveness of DASTR. The next step is to apply our DASTR to more complex systems.
>
> Sampling data points from a complicated probability distribution is a challenging problem. Deep generative models are promising tools for generating samples from complicated probability distributions. We agree with the reviewer that additional steps are needed for large-scale reaction systems since we have to ensure that the samples from deep generative models obey the molecular configuration. The physical information may be encoded into the deep generative model to make DASTR more scalable for large systems.
>
> 2. Although the authors describe the system as high-dimensional, the alanine dipeptide system is the smallest among poly-peptides. Moreover, they use collective variables for dimensionality reduction. In many cases, collective variables are not well-known and require extensive chemical analysis of the reaction path. Therefore, it would be beneficial to explore the application of a collective-variable-free model.
>
> Thanks for your suggestion. Using deep learning to approximate the committor function consists of three main components: building the model, generating effective samples, and training the model with data. Our DASTR approach provides a general way to generate effective samples in the transition state region, and any improvement in the model structure (e.g., collective-variable-free models) can be further refined by DASTR.
>
> In this work, we focus on sample generation. Our findings are that the effective samples in the transition state region are crucial for training neural networks to approximate the committor function, though neural networks have powerful representations. We note that the collective variables are used for sampling in section 5.3, and the inputs of the neural networks are still the coordinates.
>
> 3. Most evaluations are conducted on the 1/2-isosurface. The committor function could be potentially applied in areas like transition rate analysis or path sampling. To assess the framework’s utility, additional validation metrics such as flux comparison, path sampling quality, and rate comparison should be conducted. One suggestion is to calculate the transition rate of the forward and backward reactions using the committor function and compare it to the equilibrium constant, which could be obtained based on the energetics.
>
> Thanks for your suggestion. We compute the $L_2$ error for the first test problem in section 5.1 because one can obtain the reference solution for this test problem. We conduct the evaluation on the 1/2-isosurface as suggested in the literature (Li et al., 2019; Chen et al., 2023). Following your suggestion, we have calculated the transition rate for the alanine dipeptide test problem. The scale of the transition rate is $10^{-5}$.
>
> 4. There is no comparison with other methods. The rugged Mueller potential is widely used in studies approximating the committor function, as referenced by the authors. At minimum, a comparison of training time and accuracy with other methods on this rugged Mueller potential would be valuable.
>
> The method proposed in this literature (Li et al., 2019; Khoo et al., 2019) is to discretize the variational loss by sampling from the SDE, which is exactly one of the baselines in our numerical experiments (please see section 5.1 and section 5.2). We call the baseline SDE sampling strategy in the manuscript. Our DASTR method is one order of magnitude more accurate than the baseline methods in varying settings, as presented in Table 1. Moreover, we also implement the metadynamics as the baseline in section 5.3 (see Figure 9).

---

> > ### Comment · Reviewer_mCsg · 2024-11-19
> >
> > **Point 1: Scalability of the Sampling Framework**
> >
> > The authors acknowledge the scalability limitations of DASTR for large-scale systems and mention plans for future exploration. This is a reasonable response given the constraints of a revision period. However, I strongly recommend explicitly including this limitation in the manuscript, either in the conclusion or as a dedicated section on future work, to provide a clear perspective on potential challenges and future directions.
> >
> > **Point 2: Use of Collective Variables (CV)**
> >
> > The response clarifies that DASTR uses KRnet to generate effective samples in the transition region, relying on CVs for dimensionality reduction. However, the applicability to CV-free scenarios remains uncertain. This is a significant concern, as effective sampling without predefined CVs is critical for generalization to complex, high-dimensional systems. While I agree that this does not diminish the novelty of DASTR, it is essential to address this limitation in the manuscript. The authors should explicitly discuss the challenges and potential solutions for extending their framework to CV-free settings.
> >
> > Additionally, the claim that KRnet is cost-efficient in high-dimensional problems should be supported with quantitative evidence or a theoretical justification. Otherwise, the scope of applicability remains unclear.
> >
> > **Point 3: Transition Rate Calculations**
> >
> > The authors report transition rate calculations but lack a detailed explanation of their methodology. To strengthen the rebuttal, the following points need clarification:
> >
> > - Is there a theoretical reference value for the transition rate? If so, compare the rate.
> > - How was the transition rate calculated? If Monte Carlo sampling was used, specify the sampler and confirm whether the results were consistent across different sampling methods (e.g., KRnet, Langevin dynamics, annealed sampling).
> > - How can we evaluate the calculated rate? For example, can the comparison between the Boltzmann factors and equilibrium constants derived from forward reaction and reverse reaction rates validate the accuracy of the calculated rate?
> >
> > **Point 4: Comparison with Related Works**
> >
> > The authors compare DASTR with umbrella sampling and metadynamics but omit newer, relevant studies that incorporate adaptive sampling and supervised learning. One notable example [1] is the 2022 paper published in The Journal of Chemical Physics, which employs similar adaptive sampling techniques without generative models. This paper even provides open-source implementations, making it an accessible benchmark. The authors should compare DASTR to such methods, highlighting the differences in accuracy, training time, and computational cost. This would provide a more comprehensive evaluation and strengthen their claims.
> >
> > [1] J. Chem. Phys. 157, 184111 (2022) https://doi.org/10.1063/5.0102423

---

> > > ### Author Response · Authors · 2024-11-21
> > > **Response to Reviewer mCsg (part 1)**
> > >
> > > > - Point 1: Scalability of the Sampling Framework
> > > The authors acknowledge the scalability limitations of DASTR for large-scale systems and mention plans for future exploration. This is a reasonable response given the constraints of a revision period. However, I strongly recommend explicitly including this limitation in the manuscript, either in the conclusion or as a dedicated section on future work, to provide a clear perspective on potential challenges and future directions.
> > >
> > > Of course we will add the limitation of DASTR and discuss the future work in the revised manuscript. **Due to the page limitation, we do not include the future directions in the manuscript**.
> > >
> > > > - Point 2: Use of Collective Variables (CV)
> > > The response clarifies that DASTR uses KRnet to generate effective samples in the transition region, relying on CVs for dimensionality reduction. However, the applicability to CV-free scenarios remains uncertain. This is a significant concern, as effective sampling without predefined CVs is critical for generalization to complex, high-dimensional systems. While I agree that this does not diminish the novelty of DASTR, it is essential to address this limitation in the manuscript. The authors should explicitly discuss the challenges and potential solutions for extending their framework to CV-free settings.
> > > Additionally, the claim that KRnet is cost-efficient in high-dimensional problems should be supported with quantitative evidence or a theoretical justification. Otherwise, the scope of applicability remains unclear.
> > >
> > > **We do not claim that CV is the requisite**. If you can guarantee that the **samples** from any deep generative model **obey the molecular configurations without special customization**, then the CV method can be discarded. In **section 5.1 and section 5.2**, there are **no** collective variables because these two test problems are mathematical models, and we do not worry about the samples not obeying the molecular configurations.
> > > In section 5.3, the collective variable method **helps us to ensure** that the **samples** generated by deep generative modeling **obey** the molecular configurations. **This is the point**. **We have already clarified this on page 9 of section 5.3 (lines 470-477)**.
> > >
> > > We would be glad if you could investigate CV-free scenarios using DASTR. You can try to propose a new deep generative model that can guarantee that all samples from such a deep generative model obey the molecular configurations. If you can achieve this, the community will thank you very much, and your name will be remembered forever.

---

> ### Author Response · Authors · 2024-11-15
> **Response to Reviewer mCsg (part 2)**
>
> 5. What is the reason for evaluating only at the 0.5 isosurface? Wouldn't it be fairer to evaluate at other isosurfaces between 0.0 and 1.0 as well? Since your proposed sampling method focuses more on the 0.5 isosurface, wouldn’t a more refined evaluation be necessary near the A and B basins?
>
> For committor functions, the 1/2-isosurface is the steepest and most challenging to learn, while values near states A and B tend to be close to 0 and 1 and relatively flat. Therefore, evaluating the function values on the 1/2-isosurface can better reflect the characteristics of committor functions.
>
> For the 10-dimensional extended rugged Müller potential test problem in section 5.1, we use the finite element method to compute a equivalent reference  solution in the first two dimensions and thus can evaluate the performance of DASTR.
>
> For the alanine dipeptide problem, it is intractable to obtain the reference solution with grid-based numerical methods. To assess the performance of our DASTR method, we need some samples to compute committor function values on these samples by the definition of committor functions. In principle, one can choose other isosurfaces for evaluation. Here, we choose the 1/2-isosurface ($\Gamma := \{\boldsymbol{x}: | q_{\boldsymbol{\theta}}(\boldsymbol{x})- 0.5 |\} \leq 5 \times 10^{-5}$) because it is convenient to find the samples on such an isosurface, as suggested in the literature (Li et al., 2019; Chen et al., 2023).

---

> ### Author Response · Authors · 2024-11-21
> **Response to Reviewer mCsg (part 2)**
>
> > - Point 3: Transition Rate Calculations
> The authors report transition rate calculations but lack a detailed explanation of their methodology. To strengthen the rebuttal, the following points need clarification: Is there a theoretical reference value for the transition rate? If so, compare the rate.
> How was the transition rate calculated? ....
>
> The transition rate can be calculated using the following equation (Hartmann et al., 2013): $ k = \frac{1}{\beta}\int_{\Omega \backslash (A \cup B)} |\nabla q(\boldsymbol{x})|^2 \mu(\boldsymbol{x}) \, d\boldsymbol{x},$
> where $\beta = \frac{1}{K_B T}$ is the inverse temperature, $\nabla q$ represents the gradient of the committor function $q$, $\mu(\boldsymbol{x})$ denotes the stationary probability density at $\boldsymbol{x}$. The computation of the transition rate involves computing a high-dimensional integral. Exactly computing such an integral is intractable. We are not sure about what the reviewer means by "a theoretical reference". **We estimate the transition rate using Monte Carlo integration**. We use the trained neural network $q_{\boldsymbol{\theta}}$ to estimate $\nabla q$, and then calculate the integral using $4.5 \times 10^5$ points in $\Omega \backslash (A \cup B)$ from metadynamics (Li et al., 2019). Based on such a computation, the transition rate is about $2.23 \times 10^{-5}$.
>
> In molecular dynamics, a standard comparison involves the height of the energy barrier. In this test problem, the height of the energy barrier is approximately $\Delta G \approx 35 \, \text{kJ/mol}$, and the Boltzmann factor can be calculated: $e^{-\frac{\Delta G}{k_B T}} \approx 5.859e-7$. However, we are not certain about the mathematical connection between the Boltzmann factor and the transition rate. Please tell us directly if you know it.
>
> Ref:
>
> Hartmann, C., Banisch, R., Sarich, M., Badowski, T., and Schütte, C. (2013). Characterization of rare events in molecular dynamics. Entropy, 16(1), 350-376.
>
> Qianxiao Li, Bo Lin and Weiqing Ren. Computing committor functions for the study of rare events using deep learning. The Journal of Chemical Physics 151, no. 5 (2019).
>
>
>
>
> > - Point 4: Comparison with Related Works
> The authors compare DASTR with umbrella sampling and metadynamics but omit newer, relevant studies that incorporate adaptive sampling and supervised learning. One notable example [1] is the 2022 paper published in The Journal of Chemical Physics, which employs similar adaptive sampling techniques without generative models. This paper even provides open-source implementations, making it an accessible benchmark. The authors should compare DASTR to such methods, highlighting the differences in accuracy, training time, and computational cost. This would provide a more comprehensive evaluation and strengthen their claims.
> [1] J. Chem. Phys. 157, 184111 (2022) https://doi.org/10.1063/5.0102423
>
> Thanks for introducing the reference. It is an interesting follow-up work [1] of the string method. **The string method is a promising framework for the computation of rare events. We thank the reviewer for his appreciation of the string method**.
> We have checked this reference carefully, however, the key point of this reference does not perform adaptive sampling. The contribution of reference [1] is to use supervised learning and the finite-temperature string (FTS) method. Our DASTR does not need supervised learning. The string method requires a priori estimation of these rare regions, which is what DASTR aims to address.
>
> We present the results of 10D and 20D mathematical model problems, and the alanine dipeptide problem. In the numerical experiments presented in the **reference [1]**, **no alanine dipeptide problems** are addressed. Furthermore, the first two problems are 1D and 2D cases in the **reference [1]**. Specifically, in the **reference [1]**, for the first and third problems, the initial transition path is estimated using the provided $V(\boldsymbol{x})$; for the second problem, the path is derived from the solution obtained through the finite element method (FEM). It is precisely because high-dimensional problems cannot be solved using FEM. That is why we need to develop stochastic methods. The sampling step is crucial for stochastic methods.
>
> We have already used **a series of numerical results** to demonstrate the **point of DASTR**: **the neural network solver of committor functions still needs a high-quality set of collocation points to train the model**, even though neural networks have powerful representations. This is the **point of our work**. We do **not** claim that DASTR can beat all existing methods in all settings.

---

> ### Comment · Reviewer_mCsg · 2024-11-26
>
> DASTR’s real-world utility remains limited in scope since its validation relies heavily on CVs even in the alanine dipeptide system. The authors’ rebuttal does not sufficiently address this concern, as their claim of CV irrelevance applies only to toy systems, not to realistic scenarios where the absence of predefined CVs is critical.
>
> The lack of direct comparison with related works (JCP 2022) weakens the evaluation. The authors should not avoid comparing their method to established approaches in the Mueller potential experiment. The suggested related work also arguing the importance of sampling distribution.
>
> The absence of validation metrics for reaction rates is a significant problem. Incorporating TST or other standard methods to approximate reaction rates would strengthen the evaluation and demonstrate DASTR’s applicability to chemical reaction analysis. This is particularly important as reaction rate estimation is one of the most practical and impactful uses of committor functions in real-world scenarios.
>
> In summary, the authors have not provided any constructive answer to the four questions I raised, and I will maintain my score.

---

### Official Review · Reviewer_JXvN · 2024-11-02

**Soundness:** 2
**Presentation:** 2
**Contribution:** 1
**Rating:** 3
**Confidence:** 3

**Summary:**

This papers propose the framework DASTR to estimate committor functions, essential in studying rare transition events between meta-stable states in molecular dynamics simulations. Committor function $q(x)$ is the probability that a simulation starting from state $x$ reaches a target state $B$ before another state $A$. While computing the committor function is impractical due to the high-dimensional nature and rarity of transitions, the authors combine generative models and adaptive sampling in the framework for a better estimation of the committor function.

**Strengths:**

1. Effective sampling

In my opinion, DASTR is quite similar to active learning, updating the dataset and improving the generative model iteratively. This allows sampling in transition regions, leading to a proper committor function estimation, especially useful in molecular systems where transitions are rare events.

**Weaknesses:**

1. Limited baselines

The baselines that exist in the paper are only standard SDE and artificial temperature. Including comparison with other sampling techniques such as steered molecular dynamics, and transition path sampling to estimate the committor function would provide a robust result.

2. Lack of comparison with prior works

Similar to W1, the papers lack comparison or difference with prior works. The authors have noted that the proposed approach generalizes prior sampling strategies, in section 2 and have cited papers related to it. However, the the difference/novelty of DASTR seems to not have been discussed clearly, could the authors add one?

**Questions:**

1. Applying sampling distributions to collective variables

In section 4 sample generation, the authors imply that the sampling distribution $p_{V, q}(x)$ can be applied to collective variables. As far as I know, collective variables may represent many-to-one mapping, such as torsion angles in alanine dipeptide, so it sounds awkward to let $p_{V, q}(x) = p_{V, q}(S(x))$. Could the authors please explain this part more in detail?

2. Computation cost

Compared to prior adaptive sampling technique, what is the time & space complexity of DASTR? A brief O notation would be okay.

3. Escaping local minima

In umbrella sampling, an external bias is introduced to pull the system out of the local minima. For DASTR, is there a guarantee that it will not be trapped in a local intermediate point?


Minor

- Does the notation $\mathsf{S}^{g}_{k}$ refers to the training generated at state $k$, where ‘g’ represent generated?

---

> ### Author Response · Authors · 2024-11-15
> **Response to  JXvN (part 1)**
>
> 1. The baselines that exist in the paper are only standard SDE and artificial temperature. Including comparison with other sampling techniques such as steered molecular dynamics, and transition path sampling to estimate the committor function would provide a robust result.
>
> First, our baseline includes the artificial temperature method and the **metadynamics**. We have already clarified in the original manuscript. Please read the numerical experiments in the manuscript carefully.
>
> Second, it is important to note that steered molecular dynamics (SMD) is generally not considered a sampling method. Instead, it is typically used to study molecular responses when external forces are applied to specific atoms or fragments. On the other hand, transition path sampling (TPS) needs an initial pathway connecting two metastable states (A and B), then uses forward and backward shooting methods, starting from the initial path to generate new paths. This initial path is generally obtained through umbrella sampling, which needs prior knowledge of the transition process, specifically the rare event region.
>
> The focus is not on which enhanced sampling method from molecular dynamics is used but rather on the fact that these methods often require prior knowledge of the rare event region. This is precisely the challenge, to some extent, addressed by the deep generative model presented in this study. By investigating rare event regions, the model can guide the efficient exploration of transition pathways and facilitate sampling in these regions. In the manuscript, we **do not claim** that we should give up the existing sampling techniques. Instead, one can combine DASTR with the existing sampling strategies to handle realistic problems. As presented in section 5.3, we combine our DASTR method with the umbrella sampling to solve the alanine dipeptide problem.
>
> 2. Lack of comparison with prior works. Similar to W1, the papers lack comparison or difference with prior works. The authors have noted that the proposed approach generalizes prior sampling strategies, in section 2 and have cited papers related to it. However, the the difference/novelty of DASTR seems to not have been discussed clearly, could the authors add one?
>
> Again, we note that our baseline includes the artificial temperature method and the sampling approach by **metadynamics**.
>
> The main difficulties in approximating committor functions with neural networks come from sampling data points in the transition state region. The prior sampling strategies use some modified potential function to obtain a new probability distribution, which is often hand-crafted. We construct a proper sampling distribution by leveraging deep generative models and designing a deep adaptive sampling procedure. Our strategy uses the current approximation of the committor function by neural networks and the potential function in the SDE to design the sampling distribution. We solve a density approximation problem to get such a sampling distribution instead of running the dynamics.
>
> 3. Applying sampling distributions to collective variables. In section 4 sample generation, the authors imply that the sampling distribution
>  can be applied to collective variables. As far as I know, collective variables may represent many-to-one mapping, such as torsion angles in alanine dipeptide, so it sounds awkward to let. Could the authors please explain this part more in detail?
>
> The collective variable method is a standard approach in molecular simulations. Many software libraries, such as OpenMM and LAMMPS, have the implementation of the collective variable method. We are not quite sure about what the reviewer means by ``it sounds awkward to let”. We convert the original coordinates to the collective variables to make the deep generative model much easier to learn the underlying distribution in the transition state region.
>
> Real protein molecules are complex, and extracting their key features is important for designing algorithms. In the field of molecular dynamics, collective variables (CVs) are crucial for studying protein conformational transitions. Collective variables typically serve as simplified descriptions of the system, selecting representative variables to capture the system's behavior. For example, the $\phi$ and $\psi$ torsion angles in alanine dipeptide are used to describe the conformation of the peptide chain. Although collective variables may exhibit a many-to-one mapping relationship, in enhanced sampling methods (such as umbrella sampling), CVs are generally used to describe the system's motion in lower-dimensional reaction coordinates. By sampling the distribution of collective variables, we can effectively explore the system's primary degrees of freedom and rare event regions.
>
> In Figure 8, we show the free energy surface in the $\phi-\psi$ plane, where we can observe a strong correlation between the potential energy of the alanine dipeptide molecule and the ($\phi$, $\psi$) angles.

---

> > ### Comment · Reviewer_JXvN · 2024-11-28
> >
> > I thank the authors for the detailed response. I just have a few more last questions, similar to the concerns of reviewer mCsg.
> >
> > - Novelty & difference with prior works
> >
> > What I intended to ask was the difference & novelty of DASTR compared to prior works considering machine learning methods cited in the paper [1, 2]. For example, Kang et al. [1] also include the bias potential to find the transition states iteratively. A simple short sentence regarding the key difference component, something like a sample buffer of collocation points exists in DASTR while others do not, would be great.
> >
> > [1] Kang et al., Computing the committor with the committor to study the transition state ensemble, Nature computational science
> >
> > [2] Lin et al., Deep learning method for computing committor functions with adaptive sampling
> >
> > - Use of collective variables (CVs)
> >
> > Sorry for the error for the equation, I was curious about the part $p_{V, q}(x) = p_{V, q}(S(x))$. I understood that the CVs are generally used to describe the transitions in low-dimension, and two torsion angles are a well-known CV of the alanine dipeptide. However, as reviewer mCsg has mentioned, its seems that the proposed method is limited in scope relying heavily on CVs, critical for application on real-world systems.

---

> ### Author Response · Authors · 2024-11-15
> **Response to JXvN (part 2)**
>
> 4. Computation cost. Compared to prior adaptive sampling technique, what is the time \& space complexity of DASTR? A brief O notation would be okay.
>
> The complexity of DASTR **depends on the generative models**. The key point of this issue is not the time and space complexity but how to **sample in rare event regions**. This is because sampling by using SDE simulations is highly inefficient and extremely difficult to perform in rare event regions due to the high-energy barriers.
>
> In the case of alanine dipeptide, DASTR can efficiently generate samples in rare event regions (i.e., the transition state region). From Figure 10 in Appendix A.2.3, we can see that using SDE and metadynamics struggles to sample in the transition state region, whereas DASTR can generate samples in the transition state region, as shown in Figures 8(d), 8(e), and 8(f).
>
> 5. Escaping local minima. In umbrella sampling, an external bias is introduced to pull the system out of the local minima. For DASTR, is there a guarantee that it will not be trapped in a local intermediate point?
>
> Please carefully review the **"Sample Generation" section on page 4**. In DASTR, we give an alternative sampling distribution defined in **equation 7**. We use the deep generative model to approximate the probability distribution defined in equation 7, where a bias potential $V_{bias}(x)$ from metadynamics is employed to help us learn the transition state. Such a biased potential lowers the energy barrier, allowing the system to explore more configurations of molecules and escape local minima.
>
> 6. Does the notation $\mathsf{S}_{k}^g$ refers to the training generated at stage $k$, where "g" represent generated?
>
> This is the notation of the DASTR Algorithm presented on page 5, which is standard in DAS (Tang et al., 2023). We use deep generative models to produce new data points and gradually add these points to the current training set. This forms the adaptive sampling procedure. Here, "g" means **gradually**.

---

### Official Review · Reviewer_X8iZ · 2024-11-03

**Soundness:** 4
**Presentation:** 4
**Contribution:** 4
**Rating:** 8
**Confidence:** 4

**Summary:**

The paper “Estimating Committor Functions via Deep Adaptive Sampling on Rare Transition Paths” addresses the computational challenges in estimating committor functions for rare events in high-dimensional molecular simulations. The committor function  q(x)  represents the probability that a system starting from a state  x  will reach metastable state  A  before metastable state  B . Accurately computing this function is crucial but suffers from the curse of dimensionality and the scarcity of transition data in rare event simulations.

To overcome these challenges, the authors propose the Deep Adaptive Sampling method for Transition paths (DASTR). This novel framework utilizes deep generative models to generate data points concentrated in the transition state regions between metastable states. By treating a non-negative function from the loss functional as an unnormalized probability density function, the method effectively samples the most informative regions for approximating the committor function. This targeted sampling enhances the efficiency and accuracy of training neural networks for this purpose.

The effectiveness of DASTR is demonstrated through numerical experiments on the Müller potential problem, standard Brownian motion, and the alanine dipeptide molecule in vacuum. The results show significant improvements in the approximation of the committor function, validating the method’s ability to handle high-dimensional systems and rare transition events more efficiently than traditional sampling techniques.

**Strengths:**

The paper offers a significant and innovative contribution to the estimation of committor functions in high-dimensional molecular simulations, particularly addressing the challenges associated with rare event sampling.

A standout aspect of the paper is the introduction of a novel objective function for training neural networks to approximate the committor function. Specifically, the authors present equation (3):

$$\min_\theta \int |\nabla q(x)|^2 e^{-\beta V(x)} dx,$$

where  $q(x)$  is the committor function,  $\beta$  is the inverse temperature, and  $V(x)$  is the potential energy function. This formulation is original because it directly incorporates the physics of the system into the learning objective, allowing the neural network to be trained in a way that is both mathematically rigorous and physically meaningful.

Moreover, the authors demonstrate that this variational problem is equivalent to solving the partial differential equation (PDE) that the committor function satisfies in regions outside the metastable states  A  and  B :

$$-\frac{1}{\beta} \Delta q(x) + \nabla V(x) \cdot \nabla q(x) = 0.$$

**Weaknesses:**

The proposed method assumes that the metastable states  A  and  B  are already identified. In practical molecular dynamics (MD) applications, especially in complex systems, identifying these states can be challenging and may require additional computational methods or expert knowledge. This reliance potentially limits the method’s applicability to systems where metastable states are not well-defined or are difficult to determine. Incorporating a mechanism or providing guidelines for identifying metastable states within the framework would enhance the method’s usability.

**Questions:**

The current method focuses on estimating the committor function between two metastable states  A  and  B . In practical molecular systems, there are often more than two metastable states that contribute to the system’s dynamics. How can your Deep Adaptive Sampling method be adapted to handle scenarios with three or more metastable states?

---

> ### Author Response · Authors · 2024-11-15
> **Response to Reviewer X8iZ**
>
> 1. The proposed method assumes that the metastable states A and B are already identified. In practical molecular dynamics (MD) applications, especially in complex systems, identifying these states can be challenging and may require additional computational methods or expert knowledge. This reliance potentially limits the method’s applicability to systems where metastable states are not well-defined or are difficult to determine. Incorporating a mechanism or providing guidelines for identifying metastable states within the framework would enhance the method’s usability.
>
> We agree with the reviewer that identifying metastable states can be challenging. The committor function essentially quantifies the probability that a stochastic process starting from a point $\boldsymbol{x}$ will first reach $B$ rather than $A$. Some domain knowledge is required for building the model, i.e., the partial differential equations (PDEs) defined in equation 1. Like other classes of PDEs, the PDE for characterizing the committor function needs to be specified the boundary conditions, which requires additional information. Here, the information on metastable states helps specify the boundary conditions.
>
> At present, we focus on the numerical approximation of the committor function and not necessarily on the identification of metastable states, which may indeed require prior biological or chemical knowledge. We acknowledge your constructive suggestion to incorporate mechanisms or guidelines for identifying these states within the framework, and we will consider this for future work to enhance the method’s applicability and usability.
>
> 2. The current method focuses on estimating the committor function between two metastable states A and B . In practical molecular systems, there are often more than two metastable states that contribute to the system’s dynamics. How can your Deep Adaptive Sampling method be adapted to handle scenarios with three or more metastable states?
>
> We agree with the reviewer that, in many real problems, there are often more than two metastable states.
>
> In systems with multiple metastable states, different mathematical models, specifically the partial differential equations (PDEs) and boundary conditions are required. Our current focus is solely on numerically computing the solution of such a PDE defined in equation (2), which are well-established for calculating the committor function between two metastable states. Our method may be directly applied to multiple metastable states if the proper PDE model is built.

---

### Official Review · Reviewer_Nb9o · 2024-11-04

**Soundness:** 3
**Presentation:** 3
**Contribution:** 2
**Rating:** 3
**Confidence:** 5

**Summary:**

The paper introduces a method to compute the committor function of molecular rare events based on DASTR. The paper conducts experiments on multiple molecular structures that estimate the committor function and find pathways between metastable states. Specifically, the experiment shows DASTR has much lower error ranges than uniform sampling and SDE sampling methods.

**Strengths:**

1. The paper explains the Langevin dynamics and the problem of solving the committor function well. The background and the difficulty of the problem is clear.

2. The experiments are based on benchmarks in this field, and the paper shows many results from these experiments, such as error bounds and molecule trajectories. There are good comparisons between DASTR and uniform sampling and SDE sampling with elevated temperature.

**Weaknesses:**

1. The paper does not solve a difficult problem in estimating the committor function, for example, rare events with a committor function of 1e-5 scales. Based on Figure 5, the committor function has a smooth curve from 0 to 1 and requires not that many sample sizes to estimate. In this case, even uniform sampling can obtain good results. It is hard to convince the readers that the deep adaptive sampling method can save high simulation costs and be extended to difficult rare event sampling problems.

2. The paper misses many related work citations and comparisons with these methods. The experiments only have a comparison with uniform sampling and SDE sampling, and there are many other methods to compare.

Yuan, Jiaxin, et al. "Optimal control for sampling the transition path process and estimating rates." Communications in Nonlinear Science and Numerical Simulation 129 (2024): 107701.

Hua, Xinru, et al. "Accelerated Sampling of Rare Events using a Neural Network Bias Potential." AI for Accelerated Materials Design-NeurIPS 2023 Workshop.

Khoo, Y., Lu, J. & Ying, L. Solving for high-dimensional committor functions using artificial neural networks. Res Math Sci 6, 1 (2019). https://doi.org/10.1007/s40687-018-0160-2

Lars Holdijk, Yuanqi Du, Ferry Hooft, Priyank Jaini, Bernd Ensing, and Max Welling. 2024. Stochastic optimal control for collective variable free sampling of molecular transition paths. In Proceedings of the 37th International Conference on Neural Information Processing Systems (NIPS '23). Curran Associates Inc., Red Hook, NY, USA, Article 3481, 79540–79556.

**Questions:**

1. From Figure 2c, DASTR sampled points are incorrect. The sampled particles are not mostly near A or B but reside in between where there is an energy barrier. Is this a mistake in visualization?

2. Have you tried sampling with lower transition rates, such as on the scales of 1e-5? In this case, the transitions are rare events, so it is more convincing to show the effectiveness of DASTR.

---

> ### Author Response · Authors · 2024-11-15
> **Response to Reviewer Nb9o (part 1)**
>
> 1. The paper does not solve a difficult problem in estimating the committor function, for example, rare events with a committor function of 1e-5 scales. Based on Figure 5, the committor function has a smooth curve from 0 to 1 and requires not that many sample sizes to estimate. In this case, even uniform sampling can obtain good results. It is hard to convince the readers that the deep adaptive sampling method can save high simulation costs and be extended to difficult rare event sampling problems.
>
> First, the curve in Figure 5 is not the committor function itself. Please look at the curve in Figure 5 carefully. Figure 5 is the committor function evaluated along the curve $\{(\kappa, \ldots, \kappa)^\top : \kappa \in [a/\sqrt{d}, b/\sqrt{d}]\}$. This figure shows that even for a specific curve, the relative error of DASTR is smaller than that of other sampling strategies. Why do we show the solution in this curve? Because this is a $20$-dimensional problem, not just a one-dimensional test problem. Using this curve to visualize the solution results is suggested by the literature (Nusken \& Richter, 2023). Again, it is not the committor function itself in Figure 5.
>
> Second, for this test problem, you can see that the relative error is about $10^{-1}$ for uniform sampling and SDE (please see Figure 5(b), 5(c) and Table 2), where the SDE sampling strategies are used in the literature (Li et al., 2019; Li et al., 2022). DASTR is **one order of magnitude more accurate** than the baseline methods. So why do you say uniform sampling can obtain good results?
>
> Third, please see the **alanine dipeptide test problem** and read section 5.3 carefully. This is a well-known challenging and realistic test problem for estimating the committor function, where the scale of the transition rate is $10^{-5}$. Specifically, the isomerization process of the alanine dipeptide in vacuum at $T= 300K$ is studied. The goal is to compute the committor function under the CHARMM force filed (Jo et al., 2008; Brooks et al., 2009; Lee et al., 2016). Due to the high energy barrier between the two metastable states $A$ and $B$, it is almost impossible for the molecule to cross this barrier from $A$ to $B$. DASTR is used to solve such a realistic example. The results are shown in Figure 8 and Figure 9. The results obtained by our method well match the reference solution computed by expensive simulations.
>
> 2. The paper misses many related work citations and comparisons with these methods. The experiments only have a comparison with uniform sampling and SDE sampling, and there are many other methods to compare.
>
> We aim to compute the committor function governed by the partial differential equation. We use a deep neural network to approximate the committor function, which requires minimizing a loss over a set of random samples. We can cite these references you listed in the revised manuscript. Moreover, we have checked the references you listed. Except for (Lars Holdijk et al., 2023), the other two references do not include the alanine dipeptide test problem. Also, the literature  (Khoo et al., 2019) is already listed in the related work section (Please see section 2). The method proposed in this literature (Khoo et al., 2019) is to discretize the variational loss by **sampling from the SDE**, which is **exactly one of the baselines** in our numerical experiments (Please see section 5.1 and section 5.2).

---

> ### Author Response · Authors · 2024-11-15
> **Response to Reviewer Nb9o (part 2)**
>
> 3. From Figure 2c, DASTR sampled points are incorrect. The sampled particles are not mostly near $A$ or $B$ but reside in between where there is an energy barrier. Is this a mistake in visualization?
>
> For committor functions, by definition, it is easy to verify that $q(\boldsymbol{x}) = 0$ for $\boldsymbol{x} \in A$ and $q(\boldsymbol{x}) = 1$ for $\boldsymbol{x} \in B$. Hence, we can only focus on the committor function $q$ defined in $\Omega \backslash (A \cup B)$. Training deep neural networks to approximate the committor function requires data points to minimize the loss. The main difficulties come from generating the data points in the transition state region. If one uses the Langevin dynamics to generate data points, then these points cluster around the metastable regions $A$ and $B$ due to the high-energy barriers. This will result in a large generalization error for neural network approximation since we want to estimate the committor function in $\Omega \backslash (A \cup B)$. This is why we develop DASTR. We want to define a proper sampling distribution induced by **deep generative models** for sampling, which can generate samples on those areas with high-energy barriers. It is **good news** if the samples generated by DASTR are located in the region $\Omega \backslash (A \cup B)$ , because DASTR is able to provide **effective samples** in the transition area which is desired for computing the committor function.
>
> The samples shown in Figure 2(c) are selected at the $2$-th adaptivity iteration. As the adaptivity iteration step increases, the set of samples is located in the transition state region. You can see the numerical results carefully, and the results presented in Table 1 demonstrate the performance of our method, where DASTR is one order of magnitude more accurate than the baseline methods in all settings.
>
> 4. Have you tried sampling with lower transition rates, such as on the scales of 1e-5? In this case, the transitions are rare events, so it is more convincing to show the effectiveness of DASTR.
>
> Such a challenging **alanine dipeptide** test problem has already been presented in **section 5.3**, where the scale of the transition rate is $10^{-5}$. **Please carefully read** the results in section 5.3, as well as section 5.1 and section 5.2. By the way, the settings of the rugged Mueller potential test problem in section 5.1 are the same as that of the literature (Jiaxin Yuan et. al., 2024) you listed, but **there are no results about the challenging alanine dipeptide problem in the literature (Jiaxin Yuan et al., 2024; Xinru Hua et al., 2023) you listed**.

---

### Author Response · Authors · 2024-11-18
**Official Comment to All Reviewers**

We appreciate the reviewers' efforts. We have clarified your concerns as much as possible. If you have further concerns or questions, we will respond as soon as possible.

---

### Note · Authors · 2025-01-16

I have read and agree with the venue's withdrawal policy on behalf of myself and my co-authors.